# Novel Fe_3_O_4_ Nanoparticles with Bioactive Glass–Naproxen Coating: Synthesis, Characterization, and In Vitro Evaluation of Bioactivity

**DOI:** 10.3390/ijms25084270

**Published:** 2024-04-12

**Authors:** Thalita Marcolan Valverde, Viviane Martins Rebello dos Santos, Pedro Igor Macário Viana, Guilherme Mattos Jardim Costa, Alfredo Miranda de Goes, Lucas Resende Dutra Sousa, Viviane Flores Xavier, Paula Melo de Abreu Vieira, Daniel de Lima Silva, Rosana Zacarias Domingues, José Maria da Fonte Ferreira, Ângela Leão Andrade

**Affiliations:** 1Departamento de Morfologia, Instituto de Ciências Biológicas (ICB), Universidade Federal de Minas Gerais (UFMG), Belo Horizonte 31270-901, MG, Brazil; thalitamarcolan@gmail.com (T.M.V.); macarioviana13@gmail.com (P.I.M.V.); costagmj@gmail.com (G.M.J.C.); 2Departamento de Química, Instituto de Ciências Exatas e Biológicas (ICEB), Universidade Federal de Ouro Preto (UFOP), Ouro Preto 35400-000, MG, Brazil; vivianesantos@ufop.edu.br (V.M.R.d.S.); dlimadaniels@gmail.com (D.d.L.S.); 3Departamento de Patologia Geral, Instituto de Ciências Biológicas (ICB), Universidade Federal de Minas Gerais (UFMG), Belo Horizonte 31270-901, MG, Brazil; alfredomgoes@gmail.com; 4Laboratório de Fitotecnologia, Escola de Farmácia, Programa de Pós-Graduação em Ciências Farmacêuticas, Universidade Federal de Ouro Preto (UFOP), Ouro Preto 35400-000, MG, Brazil; lucasresendedutrasousa@gmail.com (L.R.D.S.); viviane.xavier@aluno.ufop.edu.br (V.F.X.); 5Laboratório de Morfopatologia, Núcleo de Pesquisas em Ciências Biológicas, Universidade Federal de Ouro Preto (UFOP), Ouro Preto 35400-000, MG, Brazil; paula@ufop.edu.br; 6Departamento de Química, Instituto de Ciências Exatas (ICEx), Universidade Federal de Minas Gerais (UFMG), Belo Horizonte 31270-901, MG, Brazil; rosanazd@yahoo.com.br; 7Departamento de Engenharia de Materiais e Cerâmica, CICECO, Universidade de Aveiro (UA), 3810193 Aveiro, Portugal; jmf@ua.pt

**Keywords:** magnetic nanoparticles, silica, naproxen, biocompatibility

## Abstract

Immune response to biomaterials, which is intimately related to their surface properties, can produce chronic inflammation and fibrosis, leading to implant failure. This study investigated the development of magnetic nanoparticles coated with silica and incorporating the anti-inflammatory drug naproxen, aimed at multifunctional biomedical applications. The synthesized nanoparticles were characterized using various techniques that confirmed the presence of magnetite and the formation of a silica-rich bioactive glass (BG) layer. In vitro studies demonstrated that the nanoparticles exhibited bioactive properties, forming an apatite surface layer when immersed in simulated body fluid, and biocompatibility with bone cells, with good viability and alkaline phosphatase activity. Naproxen, either free or encapsulated, reduced nitric oxide production, an inflammatory marker, while the BG coating alone did not show anti-inflammatory effects in this study. Overall, the magnetic nanoparticles coated with BG and naproxen showed promise for biomedical applications, especially anti-inflammatory activity in macrophages and in the bone field, due to their biocompatibility, bioactivity, and osteogenic potential.

## 1. Introduction

Bioactive glasses (BGs) represent an important class of biocompatible and bioactive materials developed for bone tissue repair and regeneration. During the 1970s and 1980s, bioactive glasses were obtained by melting and rapid cooling, obtaining materials with low specific surface area and porosity [1]. On the other hand, since the early 1990s, the sol–gel method has been applied to obtain BGs with much higher specific surface area and porosity than previous glasses [2,3]. The greater surface area and porosity significantly increase the bioactivity of sol–gel-derived BGs [4].

Bioactive materials are those that can induce biological responses upon interacting with proteins, cells, or tissues in vivo [5]. In general, the responses provided by bioactive materials include the capability to bond to hard or soft tissues [6]; stimulate cell adhesion, differentiation, and proliferation [7]; mimic the bio-matrix for tissue regeneration [8]; recognize specific proteins and/or cells; release bioactive ions or molecules [9]; and have targeted drug delivery [10]. With these properties, bioactive materials show great potentials of changing cellular behaviors and functions and eliciting specific responses from living tissues for diagnostics, therapeutics, and regenerative medicine [11]. Bioactive nanomaterials are an important subclass of biomaterials. They are not simply miniaturized versions of macroscopic materials. Bioactive nanomaterials exhibit unique bioactivities due to their nanoscale size, high specific surface area, and precise nanostructure. These properties significantly influence the interactions between materials and biological systems [12]. Based on this, the development of glass-ceramics with high bioactivity (from here on, we will talk about bioactivity in relation to bonding with hard and soft tissues) and magnetic properties has also attracted much attention. These multifunctional systems are able to combine bone regeneration and drug release abilities [13,14,15,16], with enhanced mechanical properties due to added metal oxides [17,18]. However, the introduction of extra metal oxides should not alter the bioactivity of the glasses.

Moreover, bioactive and ferrimagnetic glass-ceramics are useful as thermo-seeds for hyperthermia treatment of cancer [19,20,21], especially for deep-seated cancers such as bone tumors. When implanted around tumors, they can directly bond to the tumorous bone as a result of bioactivity, which can further stimulate bone regeneration [22]. When such a material is placed in the tumor region and is subjected to an alternating magnetic field, the heat from thermo-seeds raises the temperature of the surrounding area. Magnetic bioactive glass ceramic implants can undergo the re-heating process when necessary and kill the malignant cells to prevent the tumor recurrence. Additionally, the harmful leaching of metal ions from these materials into human body fluid can be avoided, encapsulating each ferromagnetic particle by the glassy matrix [23]. After the cycling heating process for destroying the malignant cells, the bioactive shell of ferromagnetic particles can also reinforce and regenerate the weakened tumoral bone [24].

Clinical studies, including Phase II and Phase III trials, supported by previous biological and pre-clinical testing results, revealed that hyperthermia can significantly improve the outcomes when combined with radiation therapy [25,26]. Moreover, research studies have shown that magnetic (M) and bioactive glass (BG) composite (MBG) could be used as a chemotherapy drug carrier and realize the conjunction of hyperthermia and chemotherapy to potentiate an antitumor therapy effect [27].

Although several studies have already been conducted demonstrating the biocompatibility of magnetic nanoparticles coated with different compounds [5,28,29], the biocompatibility and cytotoxicity of the novel biomaterials is a key issue that should be addressed prior to pre-clinical applications [28]. It is important to note that the long-term use of biomaterials is often hampered by the inflammatory response elicited after implantation. Early adsorption of proteins on the biomaterial surface triggers the activation of innate and acquired immunity [30]. Macrophages are considered key effectors in this inflammatory response known as the foreign body reaction [31]. Their direct contact with the implant surface can induce the secretion of chemokines, growth factors, and pro-inflammatory cytokines such as interleukin 1β (IL-1β) among others [32]. Because these approaches have limited long-term stability, more efficient methods to suppress inflammation are based on the immobilization of drugs with anti-inflammatory effects such as nonsteroidal anti-inflammatory drugs (NSAIDs) [30]. Naproxen is a widely known NSAID with anti-inflammatory, analgesic, and antipyretic properties of superior pharmacokinetics, and it has a lower risk of cardiovascular adverse effects compared with other NSAIDs [33]. Like other NSAIDs, naproxen causes gastritis and peptic ulceration after oral administration. To avoid these side effects, this drug has been encapsulated in different materials such as polymers, hyaluronic acid, and lipids [34,35,36]. Naproxen’s high hydrophilicity makes it difficult to incorporate into other hydrophilic multilayer systems, potentially limiting its effectiveness in certain applications [37]. Some authors have improved its solubility by reducing its particle size [38]. Reducing the size can also decrease the toxicity of naproxen [39]. In our work, naproxen was solubilized in an organic solvent and intricately bonded to the BG coating matrix.

Despite the numerous studies that have been conducted on coating magnetic nanoparticles and drug release [16] or for some specific biological purpose [40,41], to the best of our knowledge, this is the first time that magnetic nanoparticles have been coated with bioactive glass and have had naproxen linked to this material. The advantages of this material lie in the combination of its individual properties:
(i)Magnetic nanoparticles can provide mechanical strength and the ability to heat up in an alternating current magnetic field.(ii)The greater surface of bioactive glass nanoparticles presents an incomparable and promising feature similar to the biological apatite. Nanoparticles improve cellular adhesion, enhance osteoblast proliferation and differentiation, and increase biomineralization for implants [42].(iii)An anti-inflammatory drug can favor bone regeneration by avoiding severe inflammation in the cancer-treated area [43].

Therefore, this study presents, for the first time, the synergistic combination of magnetic and bioactive glass nanocomposites (sol–gel-derived glass) with naproxen. The resulting biomaterial underwent comprehensive evaluation for bioactivity, biocompatibility, anti-inflammatory potential, investigation of cellular internalization, and osteogenic capability (Figure 1).

## 2. Results and Discussion

### 2.1. Characterization of the Synthesized Magnetic Nanoparticles

Figure 2a shows the high-resolution transmission electron microscopy (HRTEM) images of the iron oxide nanoparticles (mag0) sample. The selected area electron diffraction (SAD) data from this sample confirm the magnetite phase (Figure 2b). The experimental SAD patterns were compared with simulated electron diffraction profiles for standard magnetite (a = 8.3967 Å) and maghemite (a = 8.33 Å) (Figure 2c), both with a centric setting space group, *Fd-3 m. For such simulations, an acceleration voltage of 200 kV, a 6 nm crystal, and Lorentzian model for line shape were chosen. The results are in agreement with the standard magnetite sample used as a reference.

Figure 3 shows the transmission electron microscopy (TEM) micrographs of mag0, mag1 (iron oxide nanoparticles coated with tetramethylammonium hydroxide), and MBG samples (sample mag1 coated with glass). The mean diameters calculated for the mag0 and mag1 samples from several TEM images using Image J software v 1.51r were 7.0 ± 0.3 nm and 6.1 ± 0.1 nm, respectively. This difference was actually expected, as tetramethylammonium hydroxide (TMAOH) acts as a surface-active agent [20] favoring the dispersion of Fe_3_O_4_ (iron oxide) nanoparticles. Coating the nanoparticles with BG (MBG sample) was accompanied by the formation of sub-micrometer agglomerates, likely due to the attractive magnetite interactions, not disturbed by the deposition process [44].

Figure 4 shows the scanning electron microscopy (SEM) micrographs of the MBG samples before (MBG0) and after being soaked in simulated body fluid (SBF) for 7 and 14 days (MBG7 and MBG14, respectively). The surface of the MBG0 sample (Figure 4a) exhibits irregularly shaped particles of different size. The energy-dispersive X-ray spectrometry (EDS) results were obtained in point scanning. These spectra reveal that the surface is composed of Ca, P, Si, and Fe, as expected. Increasing immersing times in SBF (MBG7, MBG14) resulted in gradual but noticeable surface morphological changes, accompanied by concomitant surface enrichments in P (Figure 4b,c). Moreover, the EDS peaks relative to other elements, including Si, Fe, and O, became gradually less intense in comparison to that of MBG0 sample. These results are consistent with the formation of a surface calcium phosphate (CaP) layer as predicted by Hench [45]. Covering the surface of the naked material with a growing CaP-rich layer explains its hidden effect towards the other elements, as observed in the EDS spectra of MBG7 and MBG14.

The changes as a function of soaking time in SBF were also assessed by attenuated total reflection Fourier transform infrared (ATR-FTIR) spectroscopy measurements. The spectra of mag0 (Figure 5 (a)) shows a band at 570 cm^−1^ that corresponds to Fe–O bond vibrations in the crystalline lattice of magnetite [46]. The ATR-FTIR spectrum of the TMAOH (Figure 5 (b)) shows a strong band at 950 cm^−1^ attributed to the υ_asym_(C–N) mode, which is generally observed within the 900–1000 cm^−1^ domain [47]. The ATR-FTIR spectrum of mag1 (Figure 5 (c)) also shows a band at 550 cm^−1^ that is characteristic of magnetite, as well as the transmittance bands typical of TMAOH, proving that the peptizing agent has been strongly adsorbed at the surface of the nanoparticles.

The ATR-FTIR spectrum for the MBG sample heat-treated at 700 °C before immersion in SBF (Figure 5 (d)) shows some main vibration modes [48,49,50] that can be assigned to: (i) Si–O–Si stretching at 1000–1200 cm^−1^, (ii) non-bridging silicon–oxygen (Si–O–NBO) stretching at 870–975 cm^−1^, (iii) Si–O with non-bridging oxygen per SiO4 tetrahedron (Si–O–2NBO) band at 840 cm^−1^ [51]; and (iv) the sharp Si–O–Si or O–Si–O bending mode band around at 461 cm^−1^. The bonds identified for the groups referring to silica (Si–O–Si e Si–O^−^) will provide the formation of silanol groups, which are considered essential for the growth of the hydroxyapatite layer when the material is placed in contact with the body fluids [52].

The presence of IR bands associated with Si–O–NBO and Si–O–2NBO groups indicate that the vitreous silica network suffers a net decrease in its local symmetry by the addition of alkali and alkaline earth ions [51]. Ca^2+^ and Na^+^ ions break Si–O–Si bonds to form Si–O^−^ species present as SiO^−^–Ca^2+^–^−^OSi and SiO^−^–Na^+^–^−^OSi [53]. Generally, the network modifiers such as Ca^2+^ and Na^+^ induce the formation of non-bridging oxygens (NBOs) and, consequently, promote the dissolution of glass samples in SBF. These changes in surface chemistry contribute to the precipitation of an amorphous surface CaP layer and its eventual subsequent crystallization to hydroxyapatite [54]. This is confirmed by the ATR-FTIR spectra of the samples MBG7 and MBG14. Indeed, the bands at 1100, 604, and 563 cm^−1^ (Figure 5 (e,f)) could be assigned to phosphate groups of hydroxyapatite [55]. The band at 1100 cm^−1^ overlaps with the broad frequency asymmetric vibration band of Si–O–Si bond [56]. These results indicate that MBG sample is bioactive.

Figure 6 (a–e) shows the XRD patterns of the mag0 (a), mag1 (b), and of MBG samples before, (MBG0) (c) and after soaking in SBF for 7 days (MBG7) (d), and 14 days (MBG14) (e). The magnetite phase (Fe_3_O_4_) was identified in all samples. The most intense reflections for this phase appear at 2θ angles of 30.06, 35.45, 43.04, 53.54, 57.16, and 62.72, corresponding to the indices (2 2 0), (3 1 1), (4 0 0), (4 2 2), (5 1 1), and (4 4 0), respectively. XRD peak data of magnetite from the powder diffraction file (PDF), International Centre for Diffraction Data (ICDD) card 1-1111, was utilized for identifying the existing phase in the mag0 and mag1 samples; meanwhile, the card 19-629 showed better fits with the MBG samples before and after soaking in SBF. Calcium phosphate silicate (PDF 40-393) was identified in MBG samples before and after immersion in SBF (Figure 5 (c–e)). After soaking in SBF for 7 and 14 days, new diffraction maximums appeared. Such an increase in intensity could be attributed to the (2 1 1), (2 0 2), (2 2 2), (2 1 3), and (3 2 3) reflections of the hydroxyapatite phase (card 1-1008).

### 2.2. Drug Loading Study

ATR-FTIR is one of the extensively used techniques for the study of the solid-state interaction between drug and polymer in solid dispersions. The ATR-FTIR spectra of the samples before and after naproxen loading are shown and compared in Figure 7. In the naproxen spectrum (Figure 7 (a)), it is possible to identify: (i) a broad band at about 3200 cm^−1^ assigned to the –OH stretching of –COOH group; (ii) the characteristic absorption peaks at 1681 and 1725 cm^−1^, which are assigned to the carbonyl stretch of the carboxylic acid group, where the former wavenumber represents the hydrogen-bonded carbonyl group and the latter the free carbonyl group; and (iii) a peak due to aromatic stretching positioned at 1604 cm^−1^ [57,58,59].

The MBG sample (Figure 7 (b)) shows a peak centered at 3400 cm^−1^ that is assigned to the hydroxyl stretching vibrations of the self-associated silanol groups, and the width of the peak reflects the wide frequency distribution of the hydrogen-bonded –OH groups [60]. The absorbed molecular water in the sample is also detected by the presence of a broad peak located between 3350 and 3500 cm^−1^ and another band in the region of 1620–1640 cm^−1^ [61]. With the addition of naproxen (Figure 7 (c)), the MBG-naproxen sample, the peak broadened and shifted, indicating the formation of hydrogen bond interactions between silanol hydroxyls (Si–OH) of the silica network and carbonyls of the drug. The free acid peak also shifted to 1730 cm^−1^. This observation indicates that intermolecular hydrogen bonds are formed between the carbonyls of naproxen and the silanol hydroxyls of the silicon oxide network.

The vibration due to the hydrogen-bonded carbonyl group of naproxen might have been masked by the intense peaks of the aromatic skeleton stretching of naproxen appearing around the same position and by vibration due to the –OH of the water. Generally, hydrogen bonding causes a shift to lower wave numbers of the vibration peaks. However, the phenomenon depends on the self-association and interaction between molecules [62].

Because the carbonyl stretching region was more sensitive to the consequences of BG–naproxen interactions, Figure 8 shows the ATR-FTIR spectra of the naproxen and MBG and MBG–naproxen samples in the wavenumber range of 1600 to 1800 cm^−1^. The peaks observed in this region are ascribed to the stretching carbonyl (C=O) of naproxen.

Therefore, ATR-FTIR indicates that there was an incorporation of naproxen due to the presence of drug bands in the MBG–naproxen sample.

### 2.3. Biological Assays

#### 2.3.1. Cell Viability Assay with RAW 264.7 Cells, MC3T3-E1, and Saos-2

To estimate the safety of drug incorporation (MBG–naproxen sample), its cell viability was determined (Figure 9a–f). This type of assay is widely used to predict the safety of samples [63]. MBG and free naproxen were used as controls, and the results obtained were compared with the ISO10993-5 standard [64], which states that a substance is considered cytotoxic when cell viability is less than 70%. After 24 h of exposure of the RAW 264.7 cells to the different concentrations of the sample and standards (Figure 9a), none of the materials tested showed toxicity at the evaluated concentrations, indicating safety at this time point. On the other hand, 48 h after the treatments in RAW 264.7 cells (Figure 9b), we observed cytotoxicity at the highest concentration evaluated (100 µg.mL^−1^) in the case of the three materials, but mainly for MBG, as naproxen and MBG-naproxen were found to have a 70% cell viability interface at this time point.

The cytotoxicity of the MBG sample in our study, at the highest concentration and 48 h time point, can be explained due to the ability of macrophages to recognize and phagocytose particles as foreign bodies [65]. This recognition is even easier the larger the particle sizes, which may have led to the toxicity of this sample at the highest concentration for macrophages [66]. However, no similar effect was observed in the MBG-naproxen sample, despite its similar physicochemical characteristics to MBG. The presence of naproxen may have reduced the recognition of particles by macrophages, as it is a known non-steroidal anti-inflammatory drug that reduces or suppresses the release of interleukin-6 (IL-6). This could have decreased the pro-inflammatory effect and, consequently, the cytotoxicity [67].

Given the safe performance observed in macrophage cultures with MBG-naproxen, the next assay aimed to assess cytotoxic activity in healthy and tumor bone cells, aiming to enhance the understanding of the therapeutic potential of the samples in scenarios involving osteogenesis, tumor microenvironment, and bone repair. To achieve this goal, we employed MC3T3-E1 cells, murine pre-osteoblasts, and Saos-2, a human osteosarcoma cell line, in this study. The results of the cytotoxicity assay of MC3T3-E1 cells revealed a cell viability above 82% at different sample concentrations after 24 h of exposure, suggesting that, under these conditions, MBG-naproxen does not affect cellular functions during this period (Figure 9c). However, after 48 h of exposure, a decrease in cell viability was observed in both MBG and MBG-naproxen samples, although it was still within acceptable limits (Figure 9d). This decrease was particularly evident near the cytotoxic threshold of 70%, even at the highest evaluated doses, i.e., 50 and 100 μg.mL^−1^. Therefore, the results indicate that prolonged exposure of MC3T3-E1 cells to high doses of MBG-naproxen may interfere with crucial cellular processes, highlighting the importance of considering the use of safer doses. This may include the accumulation of toxic metabolites, which can overload intracellular detoxification components in the cytoplasm and lead to cellular dysfunction, such as oxidative stress [68]. Furthermore, some studies have investigated the effects of nanoparticle exposure on cartilage and bone cells, underscoring the importance of understanding the mechanisms by which these particles interact with cells and influence the regeneration [69,70].

When analyzing the cytotoxicity results in Saos-2 osteosarcoma cells, we found a cell viability of higher than 88% at different concentrations of the samples both after 24 and 48 h of exposure (Figure 9e,f). These findings contrasted with those observed in MC3T3-E1 cells, where cell viability decreased with exposure time and administered doses. The resistance of osteosarcoma cells to MBG-naproxen could be attributed to several specific characteristics of these tumor cells. Previous studies have demonstrated that tumor cells often exhibit altered signaling pathways, conferring resistance to cytotoxic agents, and they also tend to have a reduced expression of characteristic markers and a higher metabolic rate than normal cells, facilitating the rapid elimination of toxic substances [71,72]. Additionally, future investigations into the potential anti-inflammatory effect of naproxen on Saos-2 osteosarcoma cells, encompassing its ability to modulate the expression of inflammatory markers, the production of inflammatory mediators, and the underlying molecular mechanisms, could deepen our understanding of the interaction between naproxen and magnetic nanoparticles. Therefore, the results obtained in Saos-2 osteosarcoma cells underscore the importance of considering the specific characteristics of tumor cells when assessing the efficacy of therapeutic agents as well as the need to develop more targeted and effective treatment strategies against bone cancer.

#### 2.3.2. Nitric Oxide

The synthesis of nitric oxide (NO) by macrophages, accompanied by the formation of nitrite, is a pathological process that can occur due to the stimulation of cytokines in inflammatory processes [73]. The presence of NO can lead to the formation of highly toxic radicals that are substrates for the COX-2 enzyme, which is responsible for the production of pro-inflammatory prostaglandins [74]. In addition, NO itself has pro-inflammatory effects such as cytotoxicity, edema, and mediation of cytokine-dependent processes that can lead to tissue injury and destruction [75].

Thus, macrophages were stimulated with pro-inflammatory cytokines for an indirect assessment of NO levels and consequently their inflammation. NO levels were evaluated at two times of treatment (24 and 48 h), and untreated macrophages stimulated or not stimulated with lipopolysaccharide (LPS) + interferon γ (IFN-γ) were used as controls. Figure 10 shows the nitrite levels after 24 and 48 h of treatment with the samples. The findings indicate that MBG exhibited comparable nitrite levels with no statistically significant difference compared with both the stimulated and untreated control groups. This suggests the absence of anti-inflammatory effects of MBG at this time point, particularly when considering only the lower and medium doses due to the cytotoxicity observed in the viability assay. The presence of a control effect on the pro-inflammatory effect of NO is due to naproxen, either in free or incorporated forms, which promoted a reduction in NO levels by 50 and 100 µg.mL^−1^ (Figure 10a). In 48 h after treatments, a drop in NO levels in naproxen and MBG drug samples at the highest concentration can be observed (Figure 10b). The reduction of NO by naproxen has already been evidenced even in Wistar rats treated orally with naproxen 3 mg.kg^−1^, demonstrating the efficacy of this drug in this regard [76]. In this in vivo study, the decrease in NO is likely due to the reduction in prostaglandin production. Naproxen, a commercial drug known for inhibiting COX-1 and COX-2 [76], may be responsible for this reduction. In our work, naproxen may have inhibited nitric oxide synthase (iNOS), leading to the reduction in NO. A study using silymarin, a substance known to inhibit COX-2, demonstrated that it also inhibited the production of nitric oxide and the expression of the iNOS gene by inhibiting the activation of NF-kappaB/Rel [77]. It is possible that naproxen, which has similar effects to silymarin, also inhibits iNOS and reduces NO production. However, further studies are required to confirm this theory.

#### 2.3.3. Prussian Blue Staining for Intracellular Iron Detection

The presence of blue staining in the macrophages cells lineage, RAW 264.7 (Figure 11), shows a positive result for intracellular iron detection. Ferricyanide, which reacts with Fe^3+^ from magnetite, resulted in a bluish stain in the cytoplasm of the cells, corroborating with the phagocytosis of the two nanoparticle types MBG and MBG-naproxen. Concentrations of ferric ions were detected in all analyzed concentrations, 25 µg.mL^−1^ (first column), 50 µg.mL^−1^ (second column), and 100 µg.mL^−1^ (third column), 24 and 48 h after exposure for MBG (Figure 11a–c and Figure 11g–i, respectively) and for MBG-naproxen (Figure 11d–f and Figure 11j–l, respectively). In contrast, we can only see the counterstained with neutral red in the control group in Figure 11m,n.

#### 2.3.4. Alkaline Phosphatase Activity with MC3T3-E1 Cells

Activity of alkaline phosphatase (ALP) is regarded as an early indicator of differentiation in MC3T3-E1 cells [78]. Typically, osteoblast development progresses through three stages: proliferation, production and extracellular matrix maturation, and mineralization [79]. Bone cells follow a sequential pattern of proliferation and differentiation, culminating in the formation of calcified bone tissue resembling in vivo bone formation [80]. Hence, MC3T3-E1 cells have been extensively employed to evaluate the osteogenic potential of biomaterials and scaffolds [81,82]. In this work, ALP was evaluated during the early differentiation of pre-osteoblastic cells, and Figure 12a,b presents the results after 3 and 7 days of treatment with MBG, naproxen, and MBG-naproxen using concentrations of 25 and 50 μg.mL^−1^, considering only the lower and medium doses for this analysis given the cytotoxicity presented in the viability assay. The results revealed no statistically significant difference compared with the control in most of the analyzed groups, except for the MBG 25 μg.mL^−1^ group after 3 days. These findings align with previous studies investigating the effect of nanoparticles and BG on ALP activity in osteoblasts [83]. For instance, Ha et al. [84] reported that silica nanoparticles did not significantly affect ALP activity in osteoblasts at concentrations similar to those used in this study. Similarly, Queiroz et al. [85] examined the effect of bioactive glass on bone cells and found no significant changes in ALP activity. Additionally, studies investigating the impact of magnetic nanoparticles on ALP activity have shown similar results. For example, Panseri et al. [86] demonstrated that magnetic nanoparticles did not adversely affect ALP activity in osteoblasts, even in cultures subjected to a magnetic field.

A plausible hypothesis for the lack of increase in ALP levels is that MBG-naproxen may be exerting its effects through distinct mechanisms that do not directly impact ALP activity. Although further investigation is required to fully understand the underlying mechanisms, our results suggest that, under the tested conditions, both MBG and naproxen, whether alone or in combination, did not adversely affect osteoblast differentiation or activity.

## 3. Materials and Methods

### 3.1. Materials

The chemicals used in the synthesis of silica-coated magnetic nanoparticles were iron (III) chloride hexahydrate—FeCl_3_·6H_2_O (Sigma-Aldrich, Riedel-de Haen, Søborg, Denmark), sodium sulfite—Na_2_SO_3_ (Sigma-Aldrich, Riedel-de Haen, Søborg, Denmark), tetramethylammonium hydroxide—C_4_H_13_NO·5H_2_O (TMAOH, Sigma-Aldrich, Søborg, Germany), hydrochloric acid — HCl (Sigma-Aldrich, Riedel-de Haen, Søborg, Denmark), polyvinyl alcohol—(C_2_H_4_O)_x_ (PVA, Vetec, São Paulo, Brazil), dichloromethane CH_2_Cl_2_ (Synth, Diadema, Brazil), ammonium hydroxide—NH_4_OH (Fluka, Seelze, Germany), absolute ethanol—C_2_H_5_OH (Fluka, Seelze, Germany), tetraethyl orthosilicate—Si(OC_2_H_5_)_4_ (TEOS, Fluka, Seelze, Germany), calcium nitrate tetrahydrate—Ca(NO_3_)_2_.4H_2_O (Fluka, Seelze, Germany), sodium carbonate—Na_2_CO_3_ (Fluka, Seelze, Germany), calcium hydrogen phosphate—CaHPO_4_ (Fluka, Seelze, Germany), and naproxen (Acros Organics, New Jersey, USA).

For cell culture assays, we used Roswell Park Memorial Institute 1640 medium (RPMI-1640, Sigma-Aldrich, Gaithersburg, MD, USA), alpha minimum essential medium (α-MEM, Nova Biotecnologia, São Paulo, Brazil), McCoy’s 5A medium (Sigma-Aldrich, Gaithersburg, MD, USA), fetal bovine serum (FBS, Nova Biotecnologia, São Paulo, Brazil), fetal bovine serum (FBS, Gibco^®^, Waltham, MA, USA), gentamicin (Thermo Fisher Scientific^®^, Waltham, MA, USA), penicillin-streptomycin (Pen-Strep, Sigma-Aldrich, Gaithersburg, MD, USA), phosphate-buffered saline (PBS, Gibco, USA), trypsin-EDTA (Gibco, USA), MTT (3-(4,5-Dimethylthiazol-2-yl)-2,5-Diphenyltetrazolium Bromide) (Thermo Fisher Scientific, USA), 5-Bromo-4-chloro-3-indolylphosphate/nitroblue tetrazolium (BCIP/NBT, Thermo Fisher Scientific, USA), β-glycerophosphate (Sigma-Aldrich, Gaithersburg, MD, USA), ascorbic acid (Sigma-Aldrich, Gaithersburg, MD, USA), lipopolysaccharide (LPS, Sigma-Aldrich, Gaithersburg, MD, USA), interferon γ (IFN-γ, Sigma-Aldrich, Gaithersburg, MD, USA), sodium nitrite (Sigma-Aldrich, Gaithersburg, MD, USA), TritonTM X-100 (Sigma-Aldrich, Gaithersburg, MD, USA), dimethylsulfoxide (DMSO, Synth, Brazil), sodium dodecyl sulfate (SDS, Sigma-Aldrich, Gaithersburg, MD, USA) and hydrochloric acid (HCL, Sigma-Aldrich, USA), neutral red—Neutralrot (Merck, Darmstadt, Germany), and potassium ferrocyanide (VETEC, São Paulo, Brazil).

### 3.2. Synthesis of Iron Oxide Nanoparticles (mag0 and mag-1 Samples)

Essentially, the method described by Andrade et al. [19,20] was used. In brief, sodium sulfite and ferric chloride solution were mixed and stirred until the color of the solution changed from dark yellow to light yellow. A dilute NH_4_OH solution was then added to form a black precipitate. The obtained suspension was centrifuged, and the resulting wet cake (~2 g) was labeled as “mag0”. Then, 0.5 mL of the TMAOH aqueous solution at 25% was added to each of the four centrifuge tubes to obtain homogeneous suspensions. The total 2 mL of the resulting suspension was diluted to one tenth in distilled water, and the suspension obtained was used for coating the nanoparticles with silica. A portion of this final suspension was lyophilized to render the sample labeled “mag1”.

### 3.3. Coating the Magnetic Nanoparticles with Bioactive Glass (MBG Sample)

The magnetite/BG composite sample (MBG) was prepared by adding 1 mL of the suspension of mag1 in 80 mL of absolute ethanol at room temperature (RT), followed by ultrasonic treatment for 5 min. After that, 1 mL of TEOS and 20 mL of distilled water were added. The mixture was stirred at RT for one day, followed by centrifugation at 18,000 rpm for 30 min. The supernatant was removed, and the wet cake was re-dispersed in 80 mL of ethanol, followed by ultrasonic treatment for 5 min. Subsequently, 0.5 mL of TEOS was added to prepare solution A1. Separately, 0.843 g of Ca(NO_3_)_2_.4H_2_O, 0.103 g of Na_2_CO_3_, and 0.057 g of CaHPO_4_ were dissolved in 20 mL of distilled water to prepare solution B1. Then, solution B1 was added to solution A1 under continuous stirring for 1 day at RT, followed by centrifugation at 5000 rpm for 15 min. The coated magnetic particles were collected and thoroughly washed three times with distilled water and once with acetone, then heated in the air at 5 °C min^−1^ up to 700 °C and maintained at this temperature for 3 h. This sample was labeled as MBG.

### 3.4. Preparation of Drug Loaded MBG (MBG-Naproxen Sample)

A 50 mL beaker containing 30 mL of deionized water was placed on a controlled temperature heating plate. When the temperature reached ~70 °C, 0.12 g of PVA and 0.30 g of MBG was added under slow stirring (100 rpm) until complete dissolution of the polymer to obtain the aqueous phase. In parallel, a 25 mL beaker with 6 mL of dichloromethane was placed in another heating plate, and 0.10 g of naproxen drug was added and dissolved at 40 °C to obtain the organic phase. The organic phase was transferred to a round-bottom flask coupled to a reflux system. Then, the aqueous phase was poured into the organic phase under 500 rpm stirring for 4 h at 35 °C. To maintain an adequate volume of organic phase in the medium, during this process, approximately 20 mL of dichloromethane were added 5 times (total 100 mL). Then, the suspension was placed in an oven at 40 °C for approximately 72 h to evaporate the solvents. This sample was labeled MBG-naproxen.

### 3.5. Assessment of In Vitro Bioactivity

A simulated body fluid (SBF) [87] was prepared and buffered at pH 7.4 with tris-(hydroxymethyl)-aminomethane [(CH_2_OH)_3_CNH_2_] and HCl. The bioactivity of MBG composite in vitro was tested by immersing the powders in SBF at a concentration of 1 mg.mL^−1^ at 37 °C. In all experiments, the SBF solution was completely removed and replaced by another freshly prepared solution every 48 h.

### 3.6. Characterization

Conventional transmission electron microscopy (TEM) images were used to determine the morphology and size of the MNPs. The crystallographic phase was evaluated by selected area electron diffraction (SAED) and HRTEM (high-resolution transmission electron microscopy).

For the transmission electron microscopy analysis (TEM), a drop of the suspension of the magnetite sample was placed onto a copper mesh coated with an amorphous carbon film and then dried in an evacuated desiccator. TEM micrographs were recorded on a transmission electron microscope (Hitachi, 9000 NA, Japan). The SAED patterns were interpreted with the help of JEMS software (version 3.4922U2010). A sample morphology was studied by scanning electron microscopy (SEM) (JEOL-JSM 840 A). Prior to analysis, the samples were fastened to a sample holder with the help of a double-carbon ribbon and covered with gold. Changes in the surface composition of the glass before and after soaking in SBF were demonstrated in their energy-dispersive X-ray spectrometry (EDS) analyses. The crystalline phases were determined by X-ray powder diffraction (XRD) analyses (Rigaku Geigerflex D/Max, C Series; CuKα radiation; 2θ angle range 20°–70°; step 0.02 s^−1^) after evaporation of the liquid carrier. Phases were identified by comparing the experimental X-ray patterns with standard files compiled by the International Centre for Diffraction Data. The attenuated total reflection Fourier transform infrared (ATR-FTIR) data were collected with a FT-IR model Mattson Galaxy S-7000 instrument, and we collected 32 scans at a resolution of 4 cm^−1^.

### 3.7. Biological Assays—In Vitro Studies

#### 3.7.1. Sample Preparation

The MBG, MBG-naproxen, and naproxen were subjected to UV radiation for 30 min for complete sterilization and then dispersed in 1×-PBS and homogenized for 1 min to prepare the main suspension used in the biological assays, in accordance with pre-defined concentrations for performing in vitro studies.

#### 3.7.2. RAW 264.7 Cells Culture

The murine macrophage cell line RAW 264.7 was grown in basal culture medium containing RPMI supplemented with 10% fetal bovine serum (FBS) and gentamicin. Cells were incubated in a humidified atmosphere of 5% CO_2_ at 37 °C. Subcultures were performed at a 1:3 ratio, and the culture medium was renewed every 2 to 3 days.

#### 3.7.3. MC3T3-E1 Cells Culture

MC3T3-E1 cells, obtained from the ATCC (American Type Culture Collection, Arlington, VA) and representing a subclone pre-osteoblastic cell line immortalized from the new-born mouse calvaria, were used in this study. After thawing, the cell suspension was cultured in alpha minimum essential medium (α-MEM) supplemented with 10% FBS and streptomycin antibiotics (100 μg.mL^−1^)/penicillin (500 units.mL^−1^). Cells were incubated in a humidified atmosphere of 5% CO_2_ at 37 °C. Subcultures were performed at a 1:4 ratio and the culture medium was renewed every 2 to 3 days. The cells until passage 20 were used in the experiments as osteoblasts undergoing continuous passages presented a decrease in the mineralization capacity [88].

#### 3.7.4. Saos-2 Cells Culture

Saos-2 cells, obtained from BCRJ (Banco de Células do Rio de Janeiro) and representing a cell line isolated from the bone of an osteosarcoma patient, were utilized in this study. Upon thawing, the cell suspension was cultured in McCoy’s 5A medium supplemented with 10% FBS and streptomycin antibiotics (100 μg.mL^−1^)/penicillin (500 units.mL^−1^). The cells were then incubated in a humidified atmosphere containing 5% CO_2_ at 37 °C. Sub-culturing was performed at a 1:4 ratio, and the culture medium was refreshed every 2 days.

#### 3.7.5. Cell Viability in RAW 264.7

The MBG, naproxen, and MBG-naproxen biocompatibility were assessed using the 3-(4,5-dimethylthiazol-2-yl)-2,5-diphenyltetrazolium bromide (MTT) method [89]. The MTT assay evaluates mitochondrial activity according to the conversion of the tetrazolium salt into formazan crystals. After cell growth and expansion, a total of 5 × 10^5^ cells.mL^−1^ were distributed per well in a 96-well microtiter plate, which was incubated at 37 °C with 5% CO_2_ for 24 h. The cells were treated with the samples MBG, naproxen, and MBG-naproxen dissolved in RPMI with dimethylsulfoxide 2% (DMSO) at the concentrations of 25, 50, and 100 µg.mL^−1^ for 24 and 48 h. Cell viability was evaluated after the incubation period using MTT method [89]. For this, the medium was removed and the wells were washed with RPMI. Then, 100 µL of RPMI without phenol red containing 10% FBS and 50 µL of filtered 2 mg.mL^−1^ MTT were added to the wells. The plates were covered and incubated for 4 h. After this time period, the reaction was stopped using 100 µL of DMSO, and the absorbance of the samples was read in a microplate reader (570 nm). The percentage of cell viability was determined using GraphPad Prism 6.0 software.

#### 3.7.6. Cell Viability in MC3T3-E1

The MBG, naproxen, and MBG-naproxen biocompatibility were assessed using the MTT method [89]. After cell growth and expansion, a total of 1 × 10^4^ MC3T3-E1 (per well) was seeded in 48-well plates per 24 h. The MBG, naproxen, and MBG-naproxen at the concentrations of 25, 50, and 100 µg.mL^−1^ were tested. Then, the cells were treated with final concentrations in addition to the viability control that received only 1× PBS and toxicity control that was exposed to 0.05% *v*/*v* Triton™ X-100 per 5 min.

After treatment, cell groups were evaluated at two time points (24 and 48 h). So, the medium was removed, and a solution containing 130 µL of α-MEM and 100 µL of MTT (5 mg.mL^−1^) was placed in each well. After 2 h, formazan crystals were visualized in an optical microscope and then dissolved in 130 µL of 10% of sodium dodecyl sulfate (SDS) in 0.01 mol.L^−1^ HCl. The culture plates were incubated in a 5% CO_2_ humidified atmosphere at 37 °C in all of the previously described steps. Next, the solution was centrifuged, 100 µL was transferred to a 96-well plate, and finally the optical density was measured at 595 nm. The assay was performed in biological triplicate, and the mean absorbance for each sample group was normalized by the viability control group mean result. The percentage of cell viability was determined using GraphPad Prism 6.0 software.

#### 3.7.7. Cell Viability in Saos-2

The MBG, naproxen and MBG-naproxen biocompatibility were assessed using the MTT method [89]. After cell growth and expansion, a total of 1.5 × 10^4^ Saos-2 (per well) was seeded in 48-well plates per 48 h. MBG, naproxen and MBG-naproxen at concentrations of 25, 50, and 100 µg.mL^−1^ were used. Then, the cells were treated with final concentrations in addition to the viability control that received only 1× PBS and toxicity control that was exposed to 0.1% *v*/*v* Triton™ X-100 per 5 min. After treatment, cells groups were evaluated at the end of two time points (24 and 48 h). So, the medium was removed, and solution containing 130 µL of McCoy’s medium and 100 µL of MTT (5 mg.mL^−1^) was placed in each well. After 2 h, formazan crystals were visualized in an optical microscope and then dissolved in 130 µL of 10% of SDS in 0.01 mol.L^−1^ HCl. The culture plates were incubated in a 5% CO_2_ humidified atmosphere at 37 °C in all of the previously described steps. Next, the solution was centrifuged, 100 µL was transferred to a 96-well plate and, finally, the optical density was measured at 595 nm. The assay was performed in biological triplicate, and the mean absorbance for each sample group was normalized by the viability control group mean result. The percentage of cell viability was determined using GraphPad Prism 6.0 software.

#### 3.7.8. Alkaline Phosphatase Activity

The alkaline phosphatase activity (ALP) was evaluated using the bromo-4-chloro-3-indolyl phosphate kit (BCIP) and nitro-blue tetrazolium salt (NBT) as per supplier specifications. A total of 500 MC3T3-E1 cells per well were seeded in 24-well plates per 24 h. Subsequently, the cells were treated with 25 and 50 µg.mL^−1^ of MBG, naproxen, and MBG-naproxen prepared in α-MEM culture medium supplemented with osteogenic solution (2.165 mg.mL^−1^ β-glycerophosphate + ascorbic acid). The negative control received only 1× PBS. After treatment, cell groups were evaluated at the end of 3 and 7 days. For these measurements, the supernatant was removed, and the cells were washed twice with 1× PBS. Then, 200 μL of BCIP-NBT solution was added per well and incubated at 37 °C in a 5% CO_2_ humidified atmosphere per 2 h. Subsequently, the precipitate was solubilized in 210 μL of SDS in 10% HCl. After 18 h, the well content was centrifuged, 100 μL was transferred to a 96-well plate, and the optical density was measured at 595 nm. The assay was performed in biological duplicate, and the mean absorbance for each sample group was normalized by the negative control group mean result. The percentage of alkaline phosphatase activity was determined using GraphPad Prism 6.0 software.

#### 3.7.9. Prussian Blue Staining for Intracellular Iron Detection

The Prussian blue histological staining method was utilized to detect intracellular iron concentrations [90,91]. RAW 264.7 cells in RPMI (1.5 mL) were plated into a cover glass at 3 × 10^5^ cells/well in 6-well plates. Cells were exposed to MBG and MBG-naproxen at all concentrations (25, 50, and 100 µg.mL^−1^) for 24 and 48 h. The cells were fixed in 4% paraformaldehyde for 30 min and then washed twice with 1× PBS. Prussian blue solution (equal volumes of 13% HCl and 10% aqueous solution of potassium ferrocyanide) was added to each well for 30 min. After staining, any ferric iron (Fe^3+^) present in the cells was revealed as a blue pigment. Cells were counterstained with neutral red dye for 5 min to enhance the visualization and distinction between cellular structures and iron deposits. The results were visualized with a compound light microscope (Olympus BX40 microscope) and registered with an Olympus DP25 microscope camera in the magnification of 1000×.

#### 3.7.10. Nitric Oxide

Macrophages of RAW 264.7 were distributed in 96-well plates (5 × 10^5^ cell) and then incubated at 37 °C with 5% of CO_2_ for 24 h. After, they were separately treated with the samples MBG, naproxen, and MBG-naproxen dissolved in 2% DMSO (25, 50, and 100 µg.mL^−1^) and stimulated or not stimulated with lipopolysaccharide (1.25 μg.mL^−1^) and IFN-γ (5 ng.mL^−1^). After incubation for 24 h, the supernatant was removed and stored at −80 °C for analysis of nitric oxide (NO). The experiment was performed in triplicate. NO was analyzed indirectly by the quantification of nitrite by the Griess reaction method [92]. Nitrite concentrations were determined by extrapolation from the standard curve, constructed using various concentrations of sodium nitrite, and the results were expressed as nanomolars (nmol.L^−1^). Absorbance values were measured using a microplate reader at 570 nm. Statistical differences were evaluated using GraphPad Prism 6.0 software.

#### 3.7.11. Statistical Analysis

The normality of the in vitro data was initially assessed, followed by statistical analysis using a two-way ANOVA test for cell viability and a one-way ANOVA test for the ALP assay. Post-tests were conducted using Tukey’s method for both analyses. All statistical analyses were performed using GraphPad Prism 6.0 software, with the reported values expressed as the mean ± SD. Results with *p* < 0.05 were considered statistically significant. Additionally, nitric oxide dosages in RAW 264.7 cells were analyzed using one-way ANOVA followed by Dunnett’s post-test, and results were reported similarly as the mean ± SD utilizing GraphPad Prism 6.0 software.

## 4. Final Discussion

This study demonstrates for the first time the successful synthesis of a novel combination involving glass-coated magnetic nanoparticles bonded to naproxen. XRD and ATR-FTIR analyses confirmed the presence of magnetite as the synthesized iron oxide phase, while ATR-FTIR data validated the effectiveness of the nanoparticle coating process. Additionally, the glass-ceramic material exhibits the ability to form a hydroxyapatite coating on its surface after immersion in simulated body fluid (SBF) for 7 days. Our cellular assays suggest a favorable safety profile for the synthesized samples. The presence of blue staining in the macrophage cell lineage indicates successful intracellular iron detection, supporting the phagocytosis of both MBG and MBG-naproxen nanoparticles. Notably, the immediate anti-inflammatory effects of naproxen are evident through the reduction of LPS + IFN-γ secretion, a master regulator of inflammation. This study represents a proof-of-concept for the development of novel surface coatings capable of attenuating macrophage inflammatory responses to biomaterials, which could potentially mitigate or alleviate inflammation following the implantation of medical devices, including soft or hard tissue implants. Furthermore, the evaluation of osteoblast and Saos-2 cell viability, coupled with alkaline phosphatase (ALP) activity assessment, provides valuable insights into the biocompatibility and osteogenic potential of the synthesized materials. Viability assays demonstrate a safe performance profile in osteoblasts, while Saos-2 cells exhibit notable resistance against the cytotoxic effects of the tested compounds. These findings suggest that the synthesized materials may exert differential effects on various cell types, underscoring the importance of considering cell-specific responses in biomedical applications. Moreover, the ALP analysis reveals no significant changes in ALP activity, indicating alternative mechanisms of action unrelated to direct ALP modulation.

## 5. Conclusions

In this work, the synthesis of magnetite nanoparticles and their coating with glass and an anti-inflammatory drug was demonstrated. The results showed that the synthesis was efficient and that the produced material is biocompatible under the studied conditions. This innovative material has been designed with potential applications in addressing bone-related conditions, such as bone cancer. Future studies will explore its suitability for various medical applications, including assessing optimal administration routes and evaluating biodistribution and accumulation within bones. Moreover, the present study contributes to expanding the knowledge on the development of biomaterials with enhanced biocompatibility and anti-inflammatory properties. The observed effects, combined with the cellular uptake of nanoparticles, highlight the multifaceted potential of the synthesized materials for diverse biomedical applications.

## Figures and Tables

**Figure 1 ijms-25-04270-f001:**
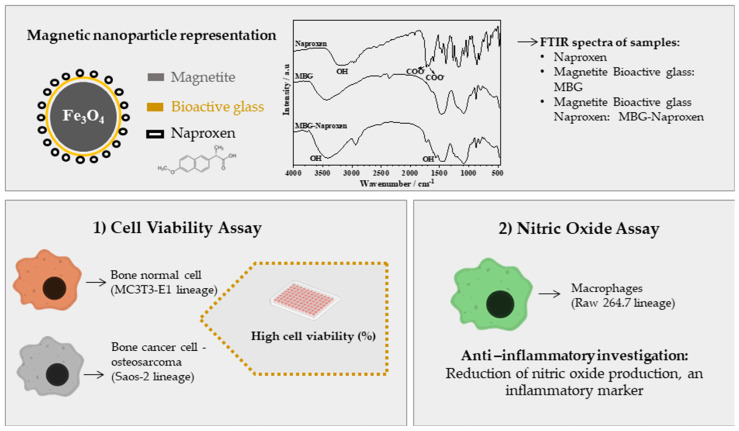
Schematics of Fe_3_O_4_ nanoparticles with bioactive glass-naproxen coating and the parameters considered in experimental design.

**Figure 2 ijms-25-04270-f002:**
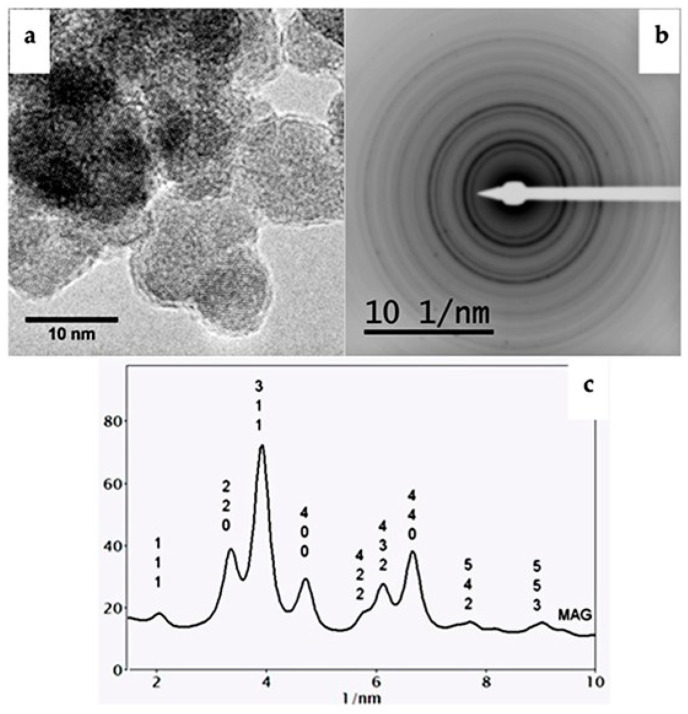
For iron oxide nanoparticles (mag0): (**a**) HRTEM images for mag0; and (**b**) SAD with inverted contrast. (**c**): Radial profiles of the SAD patterns and simulated electron diffraction profiles for standard magnetite and maghemite (both with a centric setting space group of *Fd-3 m) as a function of the reciprocal d *—spacings (dhkl* = dhkl ^−1^). The intensities for all patterns were normalized in relation to that of (113) peak reflection.

**Figure 3 ijms-25-04270-f003:**
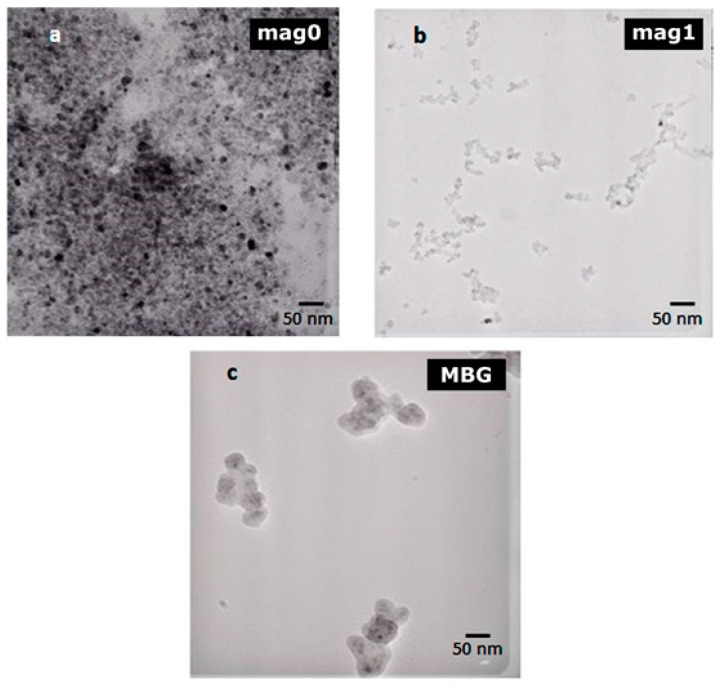
TEM micrographs of the samples: (**a**) mag0; (**b**) mag1; (**c**) MBG.

**Figure 4 ijms-25-04270-f004:**
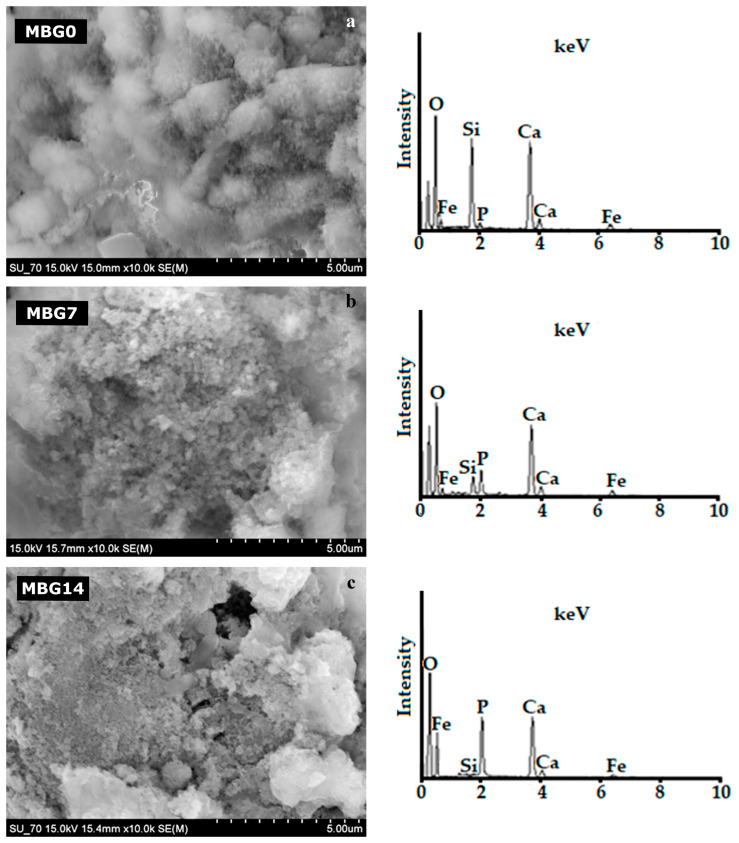
SEM micrographs of heat-treated (700 °C) MBG samples: (**a**) before (MBG0) and after being soaked in SBF for different days: (**b**) 7 days (MBG7) and (**c**) 14 days (MBG14). The insets show the respective EDS spectra obtained for the samples’ surface.

**Figure 5 ijms-25-04270-f005:**
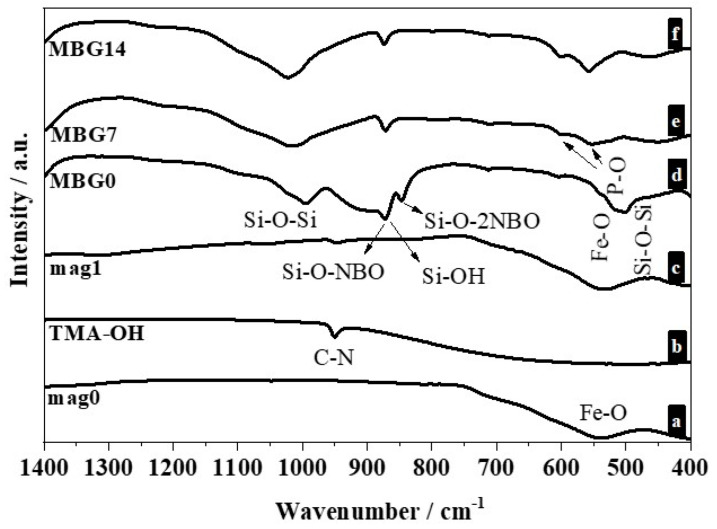
ATR-FTIR spectra of (**a**) mag0; (**b**) TMAOH; (**c**) mag1; and of MBG samples soaked in SBF for different time periods: (**d**) 0 days (MBG0); (**e**) 7 days (MBG7); and (**f**) 14 days (MBG14).

**Figure 6 ijms-25-04270-f006:**
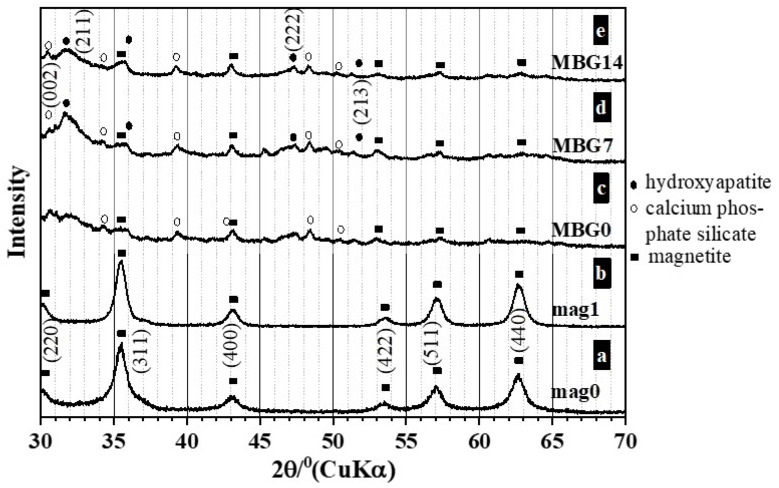
Powder X-ray diffraction patterns of: (**a**) mag0; (**b**) mag1; and of coated MBG samples before (MBG0) (**c**) and after soaking immersion in SBF for 7 days, (MBG7) (**d**) and for 14 days (MBG14) (**e**).

**Figure 7 ijms-25-04270-f007:**
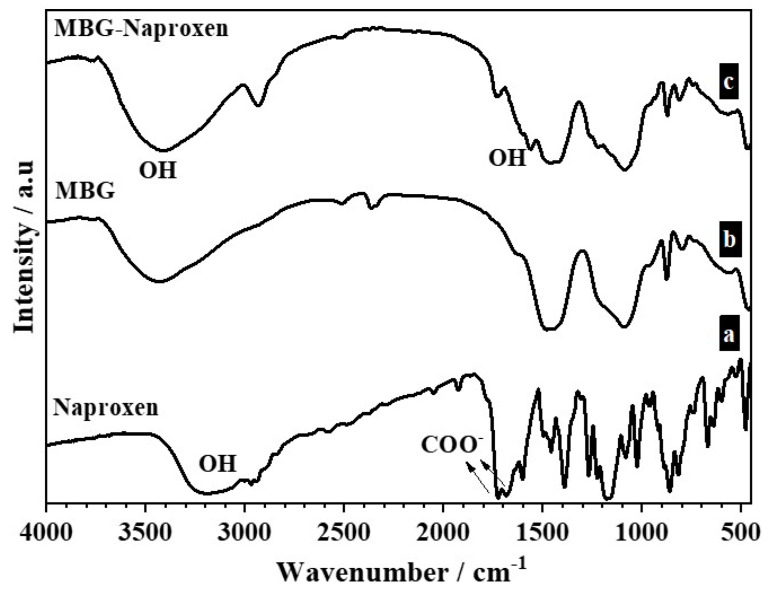
ATR-FTIR spectra of samples: (**a**) naproxen; (**b**) MBG; (**c**) MBG–naproxen.

**Figure 8 ijms-25-04270-f008:**
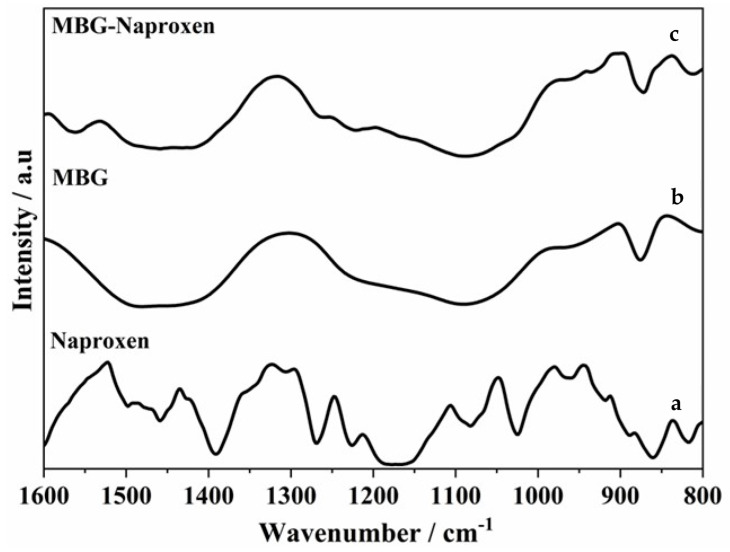
ATR-FTIR spectra of samples: (**a**) naproxen; (**b**) MBG; (**c**) MBG–naproxen.

**Figure 9 ijms-25-04270-f009:**
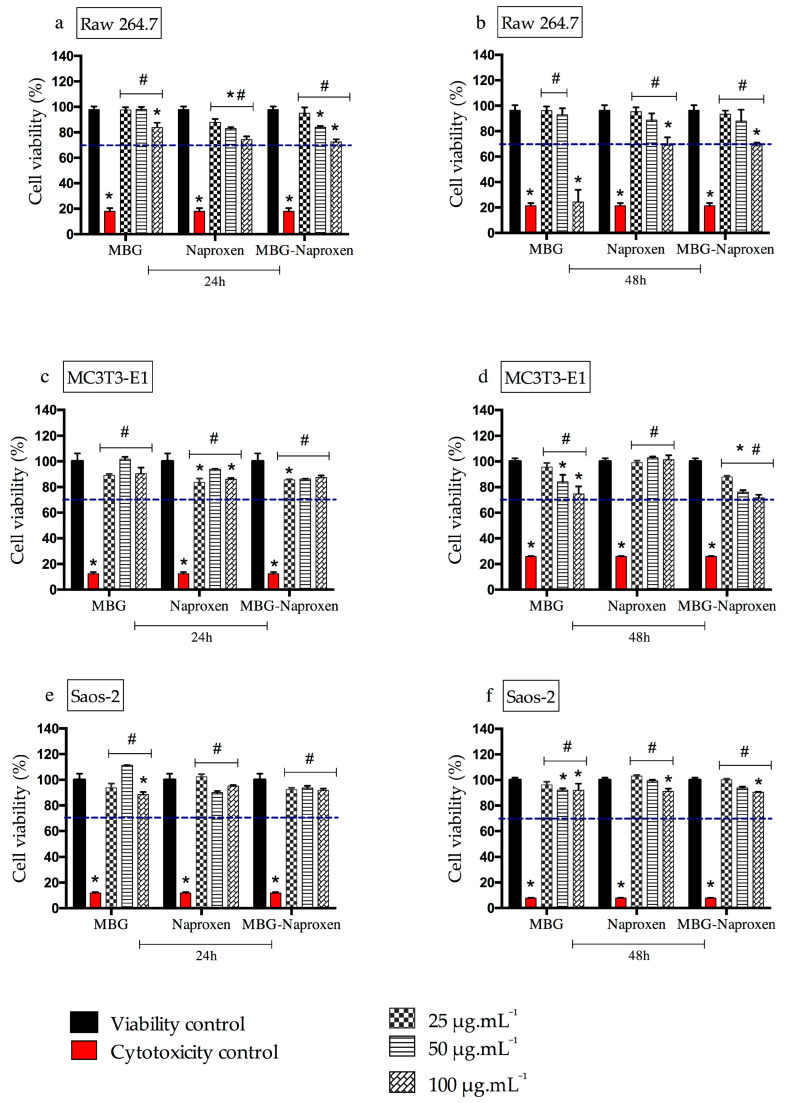
Cell viability evaluated by MTT in RAW 264.7 (**a**,**b**), MC3T3-E1 (**c**,**d**), and Saos-2 (**e**,**f**) cells exposed to MBG, naproxen, and MBG–naproxen for 24 h and 48 h at the subsequent concentrations of 25, 50, and 100 μg.mL^−1^. Results represent the mean ± SD of triplicates of the experiments, and the mean cell viability was normalized by the mean viability of the control group. The viability control used was PBS, and the cytotoxicity control used was TritonTM X-100 or DMSO 50%. (*) denotes a significant difference compared with the viability control (*p* ≤ 0.05), and (#) denotes a significant difference in relation to the cytotoxicity control (*p* ≤ 0.05) as determined by a two-way ANOVA followed by a Tukey post-test. A dashed line was used to identify the 70% viability limit according to ISO10993-5.

**Figure 10 ijms-25-04270-f010:**
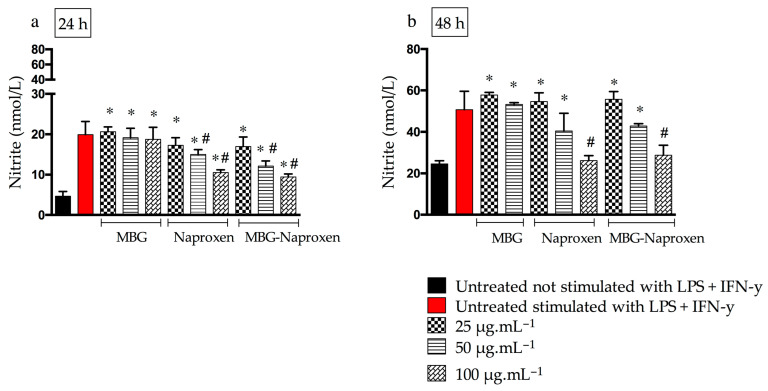
Nitrite levels evaluated in RAW 264.7 cells after 24 h (**a**) and 48 h (**b**) of MBG, naproxen and MBG–naproxen treatment at 25, 50, and 100 μg.mL^−1^. (*) Statistical difference in relation to untreated, not stimulated with LPS + IFN-γ control (*p* ≤ 0.05) and (#) statistical difference in relation to untreated, stimulated with LPS + IFN-γ control (*p* ≤ 0.05), as determined by a one-way ANOVA followed by a Dunnett’s post-test. Each bar shows the mean ± SD of triplicates of experiments.

**Figure 11 ijms-25-04270-f011:**
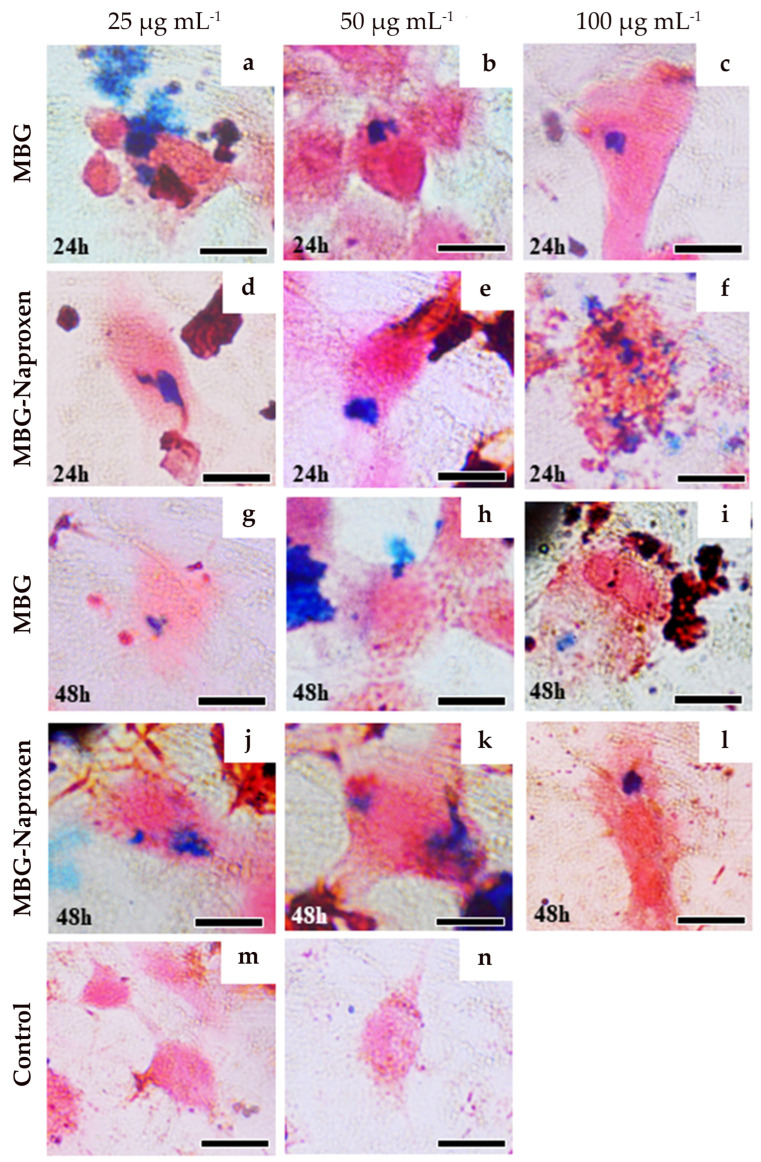
Prussian blue staining in RAW 264.7 cells for intracellular iron detection after 24 and 48 h of exposition with MBG, naproxen, and MBG–naproxen treatment at 25, 50, and 100 μg.mL^−1^. Scale bar 10 μm.

**Figure 12 ijms-25-04270-f012:**
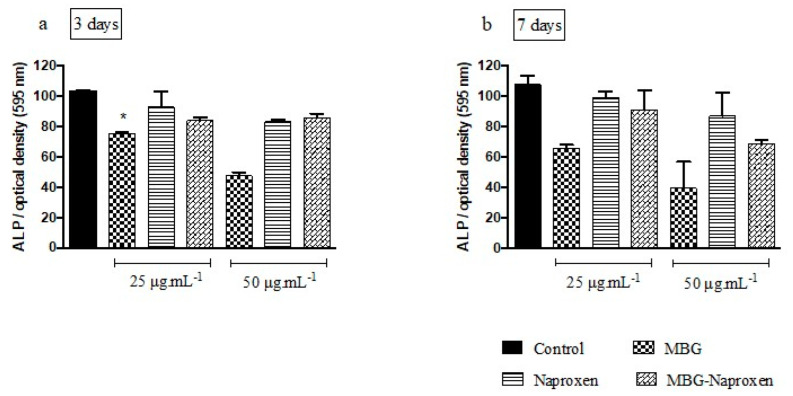
Alkaline phosphatase activity evaluated by BCIP-NBT in MC3T3-E1 cells exposed with MBG, naproxen and MBG–naproxen after 3 (**a**) and 7 days (**b**) at 25 and 50 μg.mL^−1^. Results represent the mean ± SD of duplicates of the experiments, and they were normalized by the control group (PBS). Stars denotes a significant difference compared with the control group (*p* ≤ 0.05) as determined by a one-way ANOVA followed by a Tukey post-test.

## Data Availability

Data are contained within the article.

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
