# Peer review of "Novel Fe_3_O_4_ Nanoparticles with Bioactive Glass–Naproxen Coating: Synthesis, Characterization, and In Vitro Evaluation of Bioactivity"

_ijms, 2024, doi:10.3390/ijms25084270_

Round 1
Reviewer 1 Report
Comments and Suggestions for Authors
In this manuscript, the authors reported synthesis of Fe3O4 nanoparticles with bioglass-Naproxen coating, and studied their bioactivity and anti-inflammatory performance. This work seems to be useful for this field. However, the following problems should be addressed before further consideration of publication:
1. The title can be revised to be concise, and the keywords should be reduced.
2. In the manuscript, the authors are suggested to state development and novelty of this work, especially the superiority when compared with other researches.
3. All the figures need to be revised with improved quality, where consistent layout/size are desired to improve the readability.
4. In the Introduction, the development of biological applications of Fe3O4 can be briefly revised. The recent advances should be added including: 10.1021/acsami.1c16859, 10.1016/j.colsurfa.2024.133295, 10.1002/EXP.20210089.
5. In the manuscript, a schematic illustration can be created to better show the detailed synthesis, mechanism and properties of the samples.
6. In Figure 2, were the EDS results obtained in point scanning or surface scanning mode? It should be stated clearly considering that point scanning is not reliable. SEM images of the samples can be shown in higher magnification and better quality.
7. TEM and lattice analysis should be added for nanoparticle characterization.
8. The anti-inflammatory mechanism can be analyzed in detail. Besides, the depth could be improved if the authors provided some insights of micro-/nano interactions responsible for their applications.
9. The reference format should be checked due to some errors.
Author Response
Reviewer #1
In this manuscript, the authors reported synthesis of Fe3O4 nanoparticles with bioactive glass-Naproxen coating, and studied their bioactivity and anti-inflammatory performance. This work seems to be useful for this field. However, the following problems should be addressed before further consideration of publication:
Authors´ response:
Thank you for your positive feedback on our manuscript. We made the revisions suggested to better align the research objectives.
Reviewer comment:
- The title can be revised to be concise, and the keywords should be reduced.
Authors´ response:
The title has been modified to: “Novel Fe3O4 nanoparticles with bioactive glass-Naproxen coating: Synthesis, characterization and in vitro evaluation of bioactivity”. The keywords have been reduced to: “magnetic nanoparticles; silica; naproxen; biocompatibility”.
Reviewer comment:
- In the manuscript, the authors are suggested to state development and novelty of this work, especially the superiority when compared with other researches.
Authors´ response:
The following text were added:
“Despite the numerous studies that have been conducted on coating magnetic nanoparticles and for drug release [16], or for some specific biological purpose [40,41], to the best of our knowledge, this is the first time that magnetic nanoparticles have been coated with bioactive glass and Naproxen linked to this material. The advantages of this material lie in the combination of its individual properties:
(i) Magnetic nanoparticles can provide mechanical strength and the ability to heat up in an alternating current magnetic field.
(ii) The greater surface of bioactive glass nanoparticles presents an incomparable and promising feature similar to the biological apatite. Nanoparticles improve cellular adhesion, enhance osteoblast proliferation and differentiation, and increase biomineralization for implants [42].
(iii) An anti-inflammatory drug can favor bone regeneration by avoiding severe inflammation in the cancer-treated area [43].
Reviewer comment:
- All the figures need to be revised with improved quality, where consistent layout/size are desired to improve the readability.
Authors´ response:
The resolution of the figures has been improved. They are in a separate file.
Reviewer comment:
- In the Introduction, the development of biological applications of Fe3O4 can be briefly revised. The recent advances should be added including: 10.1021/acsami.1c16859, 10.1016/j.colsurfa.2024.133295, 10.1002/EXP.20210089.
Authors´ response:
These references have been included.
Reviewer comment:
- In the manuscript, a schematic illustration can be created to better show the detailed synthesis, mechanism and properties of the samples.
Authors´ response:
An illustrative diagram has been included.
Reviewer comment:
- In Figure 2, were the EDS results obtained in point scanning or surface scanning mode? It should be stated clearly considering that point scanning is not reliable. SEM images of the samples can be shown in higher magnification and better quality.
Authors´ response:
The EDS spectra were obtained via point scanning. This information has been added. The quality of the figures has been improved.
Reviewer comment:
- TEM and lattice analysis should be added for nanoparticle characterization.
Authors´ response:
HRTEM, selected area electron diffraction (SAD), and the following text were added: “Figure 1a shows the HRTEM imagens of mag. The selected area electron diffraction (SAD) data from this sample confirm the magnetite phase. SAD patterns were further analyzed using JEMS© software (v. 3.4922U2010) (Figure 1b). The experimental SAD patterns were compared with simulated electron diffraction profiles for standard magnetite (a = 8.3967 Å) and maghemite (a = 8.33 Å) (Figure 1c), both with a centric setting space group, *Fd-3 m. For such simulations, an acceleration voltage of 200 kV, a size 6 nm crystal, and a Lorentzian model for line shape, were chosen. The results are in agreement with the standard magnetite sample used as reference.”
Reviewer comment:
- The anti-inflammatory mechanism can be analyzed in detail. Besides, the depth could be improved if the authors provided some insights of micro-/nano interactions responsible for their applications.
Authors´ response:
Thank you for the important observation. We added text with a probable anti-inflammatory mode of action (page 15, lines 441-449). Unfortunately, only with the NO dosage test it is not possible to determine the exact mode of action (please, check this article: https://pubmed.ncbi.nlm.nih.gov/12065710/). Furthermore, due to the similarity of results between free naproxen and MBG-naproxen, we prefer not to add insights of micro-/nano interactions responsible for their applications.
Reviewer comment:
- The reference format should be checked due to some errors.
Authors´ response:
The references were checked and corrected.
Reviewer 2 Report
Comments and Suggestions for Authors
The current manuscript is an interesting experimental research article on the development of Fe3O4 nanoparticles with bioglass-Naproxen coating. It appears to be overall well-structured and many relevant assays were performed. Hence, only some alterations are necessary before acceptance for publication:
- English language should be reviewed and corrected throughout the manuscript; even the title, in order to be more correct, should become something along the lines of “Novel Fe3O4 nanoparticles with bioglass-Naproxen coating: Synthesis, characterization and in vitro evaluation of bioactivity”, without the “A” in the beginning, and also erasing “anti-inflammatory activity”, since it is in itself a bioactivity, and hence it would lead to repetition;
- The innovation of the current research should be better supported, since there are already a high number of articles regarding naproxen nanoparticles, for example in:
https://pubmed.ncbi.nlm.nih.gov/35952772/
https://pubmed.ncbi.nlm.nih.gov/21130612/
https://pubmed.ncbi.nlm.nih.gov/27288817/
https://pubmed.ncbi.nlm.nih.gov/33947532/
https://pubmed.ncbi.nlm.nih.gov/32316108/
- In the introduction section, more should be said about nanosystems for therapeutic purposes, including their general characteristics (for example size and uses), advantages and disadvantages, enumeration of the different types of nanosystems that exist (inorganic and organic, and then divided into nanoparticles, micelles, liposomes, etc.), and only then does it make sense to talk about bioglass nanoparticles, their characteristics and advantages and disadvantages;
- More should be argued on the potential toxicity of bioglass nanoparticles, based on the existing literature, for example in:
https://www.ncbi.nlm.nih.gov/pmc/articles/PMC3469899/
https://www.sciencedirect.com/science/article/pii/S0928493115303283
https://www.ncbi.nlm.nih.gov/pmc/articles/PMC3813266/
https://onlinelibrary.wiley.com/doi/full/10.1002/ppsc.201800507
https://www.frontiersin.org/articles/10.3389/fchem.2019.00497/full
- In general Figure quality (resolution) should be improved;
- The “Conclusion” section is really a “Final discussion” section, so it should be renamed; additionally, a much shorter and summarized “Conclusion” section should be made and added afterwards;
- Necessary future studies should be mentioned;
- Potential administration routes and specific targeted diseases should be further discussed for the developed therapeutic system;
- An abbreviation list is missing and should be added.
Author Response
Reviewer #2
Comments and Suggestions for Authors
The current manuscript is an interesting experimental research article on the development of Fe3O4 nanoparticles with bioactive glass-Naproxen coating. It appears to be overall well-structured and many relevant assays were performed. Hence, only some alterations are necessary before acceptance for publication:
Authors´ response:
We thank the reviewer for appreciating the value of our contribution.
Reviewer comment:
- English language should be reviewed and corrected throughout the manuscript; even the title, in order to be more correct, should become something along the lines of “Novel Fe3O4 nanoparticles with bioactive glass-Naproxen coating: Synthesis, characterization and in vitro evaluation of bioactivity”, without the “A” in the beginning, and also erasing “anti-inflammatory activity”, since it is in itself a bioactivity, and hence it would lead to repetition;
Authors´ response:
Thank you very much for the comment. The change has been made.
Reviewer comment:
- The innovation of the current research should be better supported, since there are already a high number of articles regarding naproxen nanoparticles, for example in:
https://pubmed.ncbi.nlm.nih.gov/35952772/
https://pubmed.ncbi.nlm.nih.gov/21130612/
https://pubmed.ncbi.nlm.nih.gov/27288817/
https://pubmed.ncbi.nlm.nih.gov/33947532/
https://pubmed.ncbi.nlm.nih.gov/32316108/
Authors´ response:
The references have been added, and the introduction has been modified.
Reviewer comment:
- In the introduction section, more should be said about nanosystems for therapeutic purposes, including their general characteristics (for example size and uses), advantages and disadvantages, enumeration of the different types of nanosystems that exist (inorganic and organic, and then divided into nanoparticles, micelles, liposomes, etc.), and only then does it make sense to talk about bioactive glass nanoparticles, their characteristics and advantages and disadvantages;
Authors´ response:
Several references have been added, and the introduction has been modified. We believe it has improved.
Reviewer comment:
- More should be argued on the potential toxicity of bioactive glass nanoparticles, based on the existing literature, for example in:
https://www.ncbi.nlm.nih.gov/pmc/articles/PMC3469899/
https://www.sciencedirect.com/science/article/pii/S0928493115303283
https://www.ncbi.nlm.nih.gov/pmc/articles/PMC3813266/
https://onlinelibrary.wiley.com/doi/full/10.1002/ppsc.201800507
https://www.frontiersin.org/articles/10.3389/fchem.2019.00497/full
Authors´ response:
These references have been included.
Reviewer comment:
- In general Figure quality (resolution) should be improved;
Authors´ response:
The resolution of the figures has been improved. They are in a separate file.
Reviewer comment:
- The “Conclusion” section is really a “Final discussion” section, so it should be renamed; additionally, a much shorter and summarized “Conclusion” section should be made and added afterwards;
Authors´ response:
The conclusion was renamed and the following sentence was added as Conclusions. “In this work, the synthesis of magnetite nanoparticles and their coating with glass and an anti-inflammatory drug was demonstrated. The results showed that the synthesis was efficient and that the produced material is biocompatible under the studied conditions. This innovative material has been designed with potential applications in addressing bone-related conditions, such as bone cancer. Future studies will explore its suitability for various medical applications, including assessing optimal administration routes and evaluating biodistribution and accumulation within bones.”
Reviewer comment:
- Necessary future studies should be mentioned;
Authors´ response:
It was mentioned in sentence above.
Reviewer comment:
- Potential administration routes and specific targeted diseases should be further discussed for the developed therapeutic system;
Authors´ response:
A sentence has been added in the Conclusion.
Reviewer comment:
- An abbreviation list is missing and should be added.
Authors´ response:
Ok, this was done.

Round 2
Reviewer 1 Report
Comments and Suggestions for Authors
All the revisions have been checked.